

# Clinical and prognostic relevance of *CXCL12* expression in acute myeloid leukemia

Shi-sen Wang[1,*], Zi-jun Xu[2,3,*], Ye Jin[1,3], Ji-chun Ma[2,3], Pei-hui Xia[2,3], Xiangmei Wen[2,3], Zhen-wei Mao[2], Jiang Lin[2,3] and Jun Qian[1,3]

[1] Department of Hematology, Affiliated People's Hospital of Jiangsu University, Zhenjiang, Jiangsu, China
[2] Laboratory Center, Affiliated People's Hospital of Jiangsu University, Zhenjiang, Jiangsu, China
[3] Zhenjiang Clinical Research Center of Hematology, Zhenjiang, Jiangsu, China
* These authors contributed equally to this work.

Corresponding authors
Zhen-wei Mao,
maopen365@163.com
Jun Qian, qianjun@jskfhn.org.cn

## ABSTRACT

**Background:** Accumulating studies have been made to understand the association between *CXC* chemokine ligand-12 (*CXCL12*)/*CXC* chemokine receptor 4 (*CXCR4*) and acute myeloid leukemia (AML). However, large-scale data analysis of potential relationship between *CXCL12* and AML remains insufficient.

**Methods:** We collected abundant *CXCL12* expression data and AML samples from several publicly available datasets. The CIBERSORT algorithm was used to quantify immune cell fractions and the online website of STRING was utilized for gene ontology (GO) enrichment and Kyoto Encyclopedia of Genes and Genomes (KEGG) analysis. The statistical analysis and graphical work were mainly performed via the R software.

**Results:** *CXCL12* expression was extremely down-regulated in AML. Clinically, low *CXCL12* expression was correlated with higher white blood cells (WBCs) ($P < 0.0001$), more blasts in bone marrow (BM) ($P < 0.001$) and peripheral blood (PB) ($P < 0.0001$), *FLT3*-internal tandem duplications (*FLT3*-ITD) ($P = 0.010$) and *NPM1* mutations ($P = 0.015$). More importantly, reduced *CXCL12* expression predicted worse overall survival (OS) and event-free survival (EFS) in all AML, non-M3-AML, and cytogenetically normal (CN)-AML patients in three independent cohorts. As for immune cell infiltration, high *CXCL12* expressed groups tended to harbor more memory B cells and plasma cells infiltration while low *CXCL12* expressed groups exhibited more eosinophils infiltration. GO enrichment and KEGG pathways analysis revealed the potential biological progress the gene participating in.

**Conclusions:** *CXCL12* is significantly down-regulated in AML and low *CXCL12* expression is an independent and poor predictor of AML prognosis. *CXCL12* expression level correlates with clinical and immune characteristics of AML, which could provide potential assistance for treatment. Prospective studies are needed to further validate the impact of *CXCL12* expression before routine clinical application in AML.

## INTRODUCTION

Acute myeloid leukemia (AML) represents a heterogeneous disease characterizing by complex molecular signatures, complicated pathogeny and aggressive progression, as well as a poor clinical outcome (*Döhner, Weisdorf & Bloomfield, 2015*). Bone marrow (BM) microenvironment contributes significantly to leukemogenesis and leukemia progression (*Ayala et al., 2009*; *Meads, Hazlehurst & Dalton, 2008*). Most adult patients are still terribly suffering from AML, even though tremendous advancements such as novel chemotherapy drugs and hematopoietic stem cell transplantation (HSCT) have been achieved recently (*Döhner et al., 2017*; *Döhner, Weisdorf & Bloomfield, 2015*). Until now, chromosome aberration analysis remains the main method for diagnosis, classification, prognosis, and guiding clinical therapy while 40–50% of patients lack representative chromosome aberrations (*Mrózek, Heerema & Bloomfield, 2004*). Therefore, it is necessary to discover more specific and accurate biomarkers for AML. In recent years, immunophenotype and cytochemistry analysis are found valuable in the diagnosis of AML, and immunophenotyping surface type molecules like CD123, CD45, CD34, CD38 have been reported to help confirm the diagnosis of AML (*Prada-Arismendy, Arroyave & Röthlisberger, 2017*). Additionally, it is now evident that oncogene mutations affect the progression of AML and some of these mutated genes are *FLT3*, *NPM1* and *CEBPA* (*Grove & Vassiliou, 2014*; *Meyer & Levine, 2014*). Unfortunately, despite extensive researches that have been carried out to find prognostic biomarkers, AML is still a tough disease with a variable prognosis and poor survival rate, with 5-year overall survival less than 50% and even lower in the elderly (*Kell, 2004*). Considering all the above, further studies are essential for a better understanding of how various factors contribute to the disease progression.

Chemokines and their receptors mediate crucial biological functions of tumor progression including leukocyte recruitment and function, cellular senescence, tumor cell proliferation and survival, and invasion and metastasis (*Mantovani et al., 2010*). *CXC* chemokine ligand-12 (*CXCL12*), also known as stromal cell-derived factor-1 (*SDF-1*), is a *CXC* chemokine that belongs to the large family of chemotactic cytokines. The protein, functioning through its receptor *CXC* chemokine receptor 4 (*CXCR4*), plays an important role in hematopoietic cell development, immune system foundation, and retention of hematopoietic stem and progenitor cells in the BM microenvironment (*Janssens, Struyf & Proost, 2018*). *CXCL12* expression has been found to be related to at least 23 different types of tumors (*Balkwill, 2004*) and the gene's critical influences on solid tumors have been illustrated in prostate cancer (*Taichman et al., 2002*), non-small cell lung cancer (*Phillips et al., 2003*), renal cell carcinoma (*Pan et al., 2006*) and so on. As for hematological malignancies, *CXCL12* is reported to be overexpressed in chronic lymphocytic leukemia (CLL) (*Möhle et al., 1999*), as well as in childhood acute lymphoblastic leukemia (*Crazzolara et al., 2001*). Also, *Spoo et al. (2007)* have proved that elevated *CXCR4* expression is relevant to inferior prognosis of AML in adult patients (*Konoplev et al., 2007*; *Rombouts et al., 2004*; *Spoo et al., 2007*). All these findings may contribute to a better

understanding of *CXCL12*'s potential mechanism and clinical relevance in both solid tumors and blood cancers.

Despite enormous progress having been made to understand the relationship between *CXCR4* and AML, studies concentrating on *CXCL12* and AML remain insufficient, and large-scale analysis of the potential association between the gene and AML hasn't been performed before. Herein, we adopted similar methods of our previous research (*Xu et al., 2019*) and made comprehensive analyses of the gene's possible interaction with AML using public datasets, trying to reveal the implication of *CXCL12* expression in AML and providing potential treatment strategies for the clinic.

## MATERIALS & METHODS

### Patients and database

We used six publicly available datasets from Gene Expression Omnibus (GEO, http://www.ncbi.nlm.nih.gov/geo/) and The Cancer Genome Atlas (TCGA, https://cancergenome.nih.gov/) in this study. Of these datasets, three of them consist of expression data for bulk primary AML samples (the TCGA dataset, GSE6891 and GSE10358), two involve expression data for healthy and AML bone marrow samples (GSE30029 and GSE63270) and one includes expression data for major hematopoietic lineages during the differentiation (GSE42519), respectively. Expression data for FAB AML subtypes and patient characteristics were acquired from TCGA sequencing data. Normal mRNA expression data from Genotype-Tissue Expression (GTEx) database and pan-cancer cell line sequencing data from Broad Institute Cancer Cell Line Encyclopedia (CCLE) database were extracted through their portal websites for analysis.

### Estimation of immune cell fractions

The CIBERSORT algorithm (https://cibersort.stanford.edu/index.php) was used to quantify immune cell fractions according to the expression pattern of *CXCL12*. More detailed information for this method is available in Newman AM et al.'s introduction (*Newman et al., 2015*). Immune cells calculated in this study contains B cells (including B cells memory and B cells naive), T cells (including CD8+ T cells, CD4+ naive T cells, CD4+ memory resting T cells, CD4+ memory activated T cells, follicular helper T cells, regulatory T cells and T cells gamma delta), macrophage (including M0 macrophage, M1 macrophage, and M2 macrophage), NK cells (including NK cells resting and NK cells activated), dendritic cells (including dendritic cells resting and dendritic cells activated), mast cells (including mast cells resting and mast cells activated), plasma cells, monocyte, eosinophils, and neutrophils. Multiple immune infiltration deconvolution methods including TIMER, quanTIseq, xCell, MCP-counter and EPIC were achieved from their websites.

### Statistical analysis and bioinformatics

Low and high *CXCL12* expressers were discriminated against based on the median expression level of the gene. Analyses of patient characteristics and survival were described previously (*Xu et al., 2019*). Differential expression was analyzed using 'limma' or 'edgeR'

R package. The Pearson Correlation coefficient (denoted as Cor) was calculated to assess whether there was a correlation between cell infiltration and *CXCL12* expression. Moreover, 'ggplot2' and 'ggpubr' R packages were applied for visualizing the results of data analysis. The single-cell correlation analysis of *CXCL12* was conducted using single-cell data sequenced by *Van Galen et al. (2019)*, and the published scRNA-seq dataset was downloaded from GEO: GSE116256. And gene ontology (GO) enrichment and Kyoto encyclopedia of genes and genomes (KEGG) analysis were conducted using the online website of STRING (http://string-db.org). One-way ANOVA was used to test for differences among at least three groups and the Wilcoxon test was used to determine differences in each two-group comparison. All statistical analyses and figure plotting were performed using the R software version 3.5.3 (https://www.r-project.org/). All differences were considered statistically significant at the level of $P < 0.05$.

## RESULTS

### A systematic overview of differential genes and cytokines in AML

Initially, we systematically analyzed a total of 261 cytokine and cytokine receptor genes (Supplemental Data 1) in AML patients and normal controls utilizing GSE30029, and all genes were screened in Supplemental Data 2 with the difference considered significant if adjusted *p*-value < 0.05 and |logFC| > 0. As obviously displayed in Fig. 1A, the *CXCL12* gene was one of the most significantly different genes between AML patients and normal controls. Then, we further verified the differences in GSE63270 (Fig. 1B, Supplemental Data 3). The top 50 genes with significant differences were extracted from two analytic results respectively and then intersected. The common 11 genes included *CXCL12*, *TNFRSF10B*, *IL1RAP*, *IL12RB1*, *IL3RA*, *CSF2RA*, *EPOR*, *TNFSF14*, *IL18BP*, *TNFRSF4* and *IL2RG*. Furthermore, considering the essential functions of *CXCL12* in AML, we ultimately decided to study *CXCL12* in detail.

### *CXCL12* expression is different among normal tissues and various carcinomas

To explore how *CXCL12* is expressed across various tissues in healthy people, we analyzed the mRNA expression status utilizing the GTEx database (Fig. 2A, Supplemental Data 4). Apparently, *CXCL12* expression was extremely low in blood and bone marrow while it was much higher among other normal tissues. It was also worth mentioning that the brain, stomach, and testis showed a relatively low level of *CXCL12* expression compared with other tissues. More importantly, when analyzing different cancer cell lines (Supplemental Data 5), we found *CXCL12* was lowly expressed among most hematopoietic neoplasms, especially in AML and chronic myelocytic leukemia (CML). Interestingly, cervix cancer cells showed exclusively lower *CXCL12* expression levels than AML and CML, as shown in Fig. 2B.

### *CXCL12* expression is markedly reduced in AML and single cells

Considering our purpose, we studied the *CXCL12* expression difference between healthy people and AML patients. A significant decrease of *CXCL12* expression was identified in
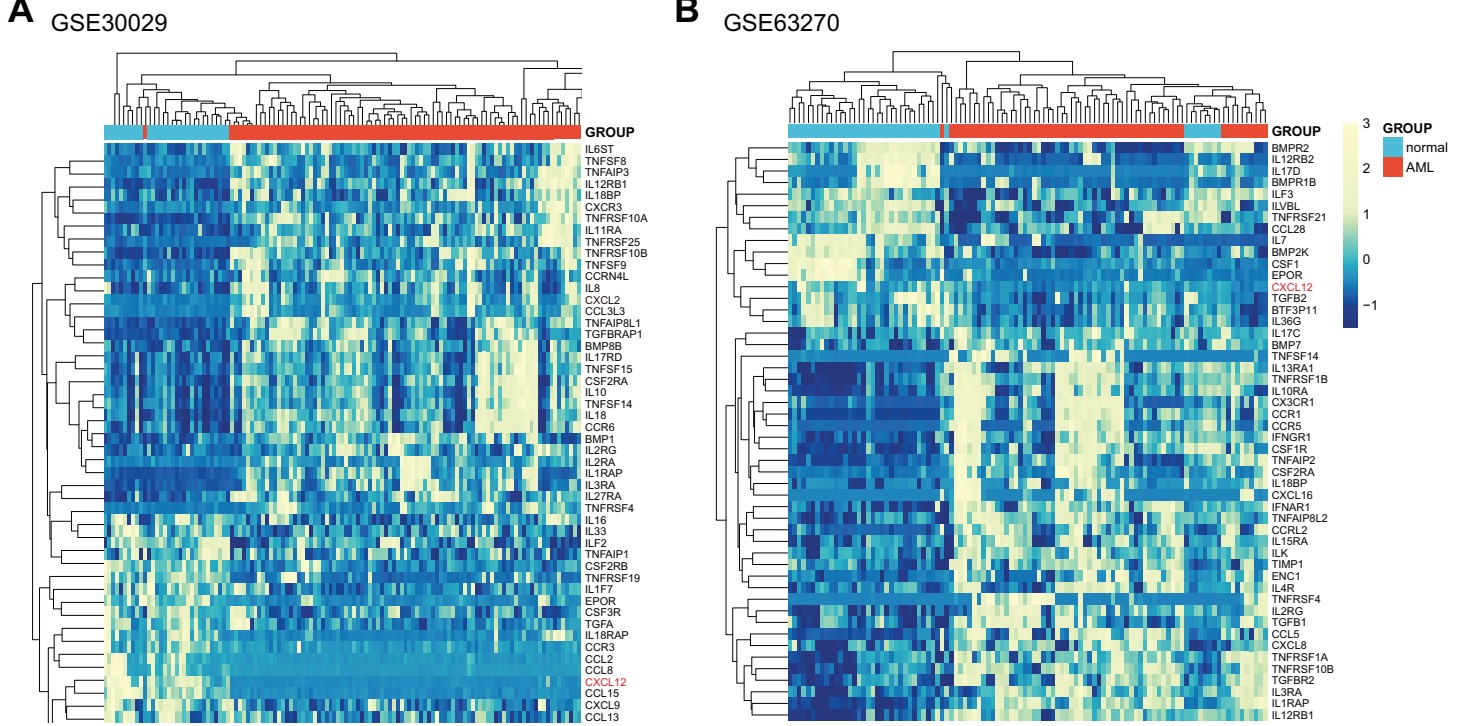

**Figure 1 Expression heat map of differentially expressed cytokine and cytokine receptor genes between AML patients and normal controls.** (A) Differentially expressed genes between AML patients and normal controls in GSE30029. (B) Differentially expressed genes between AML patients and normal controls in GSE63270. Only the top 50 genes were shown if adjusted *p*-value < 0.05 and |logFC| > 0.

AML cells (GSE30029, *n* = 90) in comparison to normal cells (GSE30029, *n* = 31) ($P$ = 3.2E−16, Fig. 3A). We detected a similar result in the GSE63270 cohort ($n_{AML}$ = 62, $n_{normal}$ = 42, $P$ = 7.4E−06), as indicated in Fig. 3B. Additionally, we analyzed *CXCL12* expression difference in eight FAB AML subtypes from M0 to M7 (Fig. S1) and *CXCL12* transcription during hematopoiesis differentiation (Fig. S2), although the difference was of no statistical significance.

For further exploration, we delineated *CXCL12* expression patterns in several hematological cells including hematopoietic stem cells (HSCs), multipotent progenitor cells (MPPs), granulo-monocyte progenitor cells (GMPs), common myeloid progenitor cells (CMPs), megakaryocyte-erythroid progenitor cells (MEPs), monocytes, and erythrocytes using the single-cell data sequenced by *Van Galen et al. (2019)*, as shown in Fig. S3A. Interestingly, unlike the former result, MPPs and MEPs showed exclusively high expression of *CXCL12*. Also, we studied *CXCL12* and *CXCR4* expression status in the single-cell sequenced data and found *CXCR4* expression was abundant in single cells from all patients while *CXCL12* was rarely expressed (Fig. S3B).

## Clinical features and molecular insights of *CXCL12*

The patient characteristics with respect to *CXCL12* expression from TCGA were summarized (Table 1). Low *CXCL12* expressed patients tended to have higher white blood cell (WBC) counts ($P$ < 0.0001) and more percentages of blasts in BM ($P$ < 0.001) and

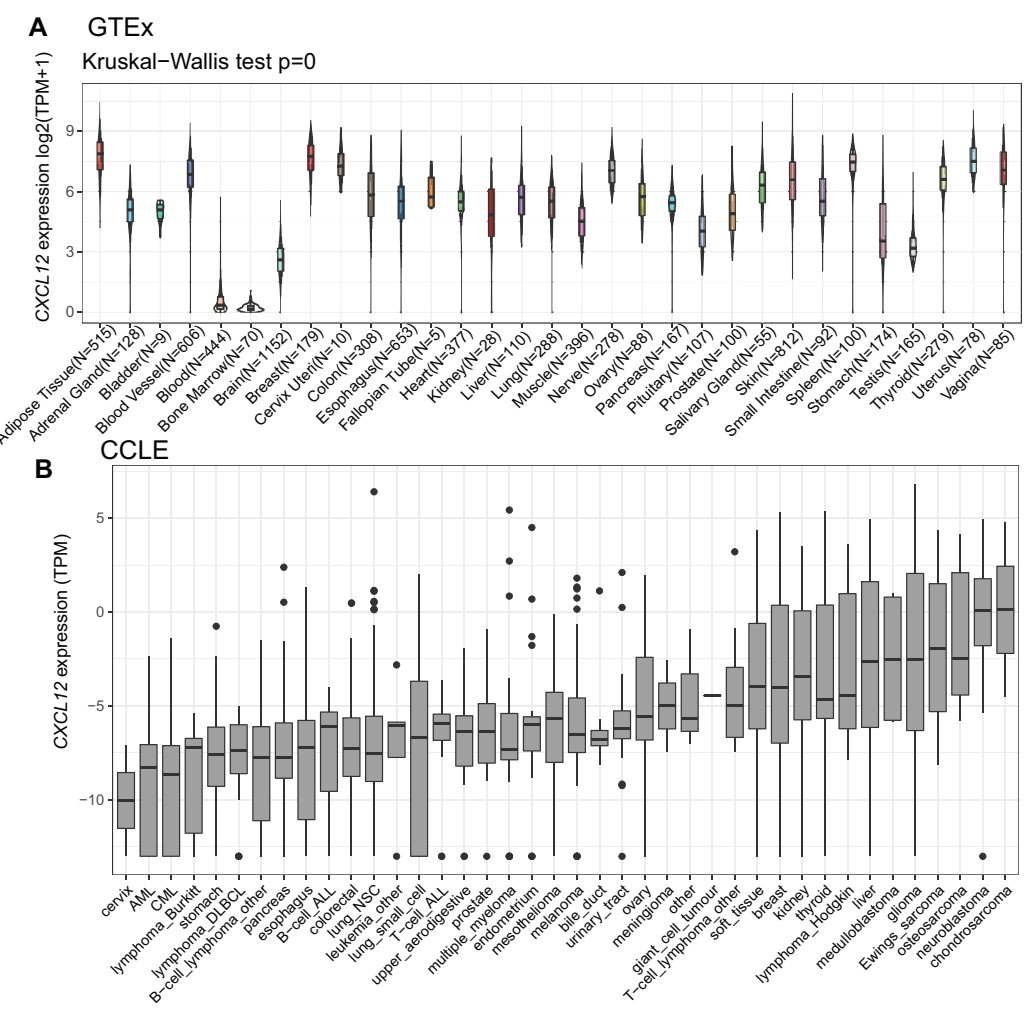

**Figure 2 mRNA expression levels of *CXCL12* from different tissue origins and tumors.** (A) mRNA expression levels of *CXCL12* in different tissues from the GTEx database. (B) mRNA expression levels of *CXCL12* in various tumor cell lines from the CCLE database.

peripheral blood (PB) (*P* < 0.0001). With regard to frequently mutated genes in AML, low *CXCL12* expressers were more likely to have *FLT3*-internal tandem duplications (*FLT3*-ITD) (*P* = 0.010) and *NPM1* mutations (*P* = 0.015) and less likely to have mutated *TP53* (*P* = 0.046) than those with high *CXCL12* expression.

## Decreased *CXCL12* expression predicts adverse prognosis

Survival analyses were further performed using three independent cohorts (TCGA, GSE6891, and GSE10358), and all cohorts were divided base on optional cut-off values determined by the x-tile method. In the TCGA whole-AML cohort, *CXCL12*[low] patients showed significantly decreased overall survival (OS) (*P* = 0.013, Fig. 4A) and event-free survival (EFS) (*P* = 0.0001, Fig. 4D). The impact of *CXCL12* expression on outcome was also observed in the non-M3-AML (OS, *P* = 0.002, Fig. 4B; EFS, *P* = 0.0007, Fig. 4E) and cytogenetically normal (CN)-AML (OS, *P* = 0.002, Fig. 4C; EFS, *P* = 0.002, Fig. 4F)

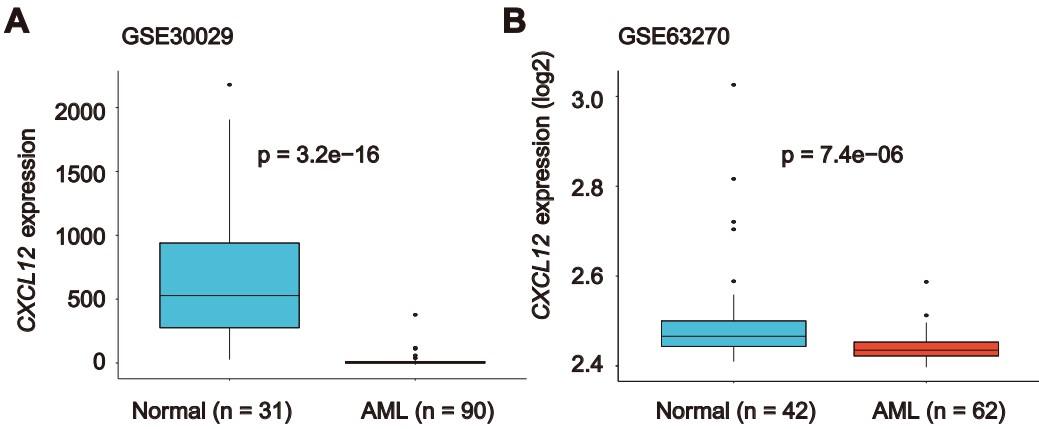

**Figure 3** **CXCL12 expression in normal hematopoiesis and AML cells.** The *CXCL12* expression level of normal controls and AML patients from GSE30029 (A) and GSE63270 (B) datasets.

subgroups. This finding was further validated using GSE6891 (Fig. S4) and GSE10358 (Fig. S5) and the results were identical to the former.

The potential relationship between low *CXCL12* expression and the existence of *FLT3*-ITD and *NPM1* mutations led us to further explore the additional prognostic value of the gene in more subdivided groups. As expected, low *CXCL12* expression patients showed significantly inferior survival probability in the *FLT3*-ITD absent (Fig. 5A, $P = 0.003$) and *NPM1* mutated (Fig. 5B, $P = 0.068$, a trend for significance actually) patients, which could provide more subdivision proofs for AML based on the dichotomous stratification of *CXCL12* expression.

## Multivariate analysis of *CXCL12* expression for OS in the TCGA cohorts

To evaluate whether there were potential interrelationships between *CXCL12* expression and known clinical factors, we first conducted a univariate analysis in the TCGA dataset and then covered variables with a univariable $P \leq 0.20$ in the multivariate Cox proportional hazard model. The multivariate analysis result of *CXCL12* expression for OS was displayed in Table 2. Low *CXCL12* expression maintained a high hazard rate for whole-AML patients (HR = 2.25, 95% CI [1.51–3.35], $P < 0.0001$), after adjusting age ($P < 0.0001$), WBC count ($P < 0.0001$), cytogenetic risk ($P = 0.002$) and mutation status of *TP53* ($P = 0.004$), *RUNX1* ($P = 0.023$) and *DNMT3A* ($P = 0.012$). The predictive significance also applied to CN-AML patients (HR = 2.14, 95% CI [1.15–3.97], $P = 0.016$), as indicated.

## Profile of relationship between *CXCL12* expression and immune cells infiltration

The landscape of the relationship between *CXCL12* expression and immune cell infiltration was systematically evaluated by the CIBERSORT algorithm (Fig. 6). We found that B cells memory, plasma cells, T cells regulatory (Tregs), T cells gamma delta, NK cells

**Table 1 Patient characteristics with respect to *CXCL12* expression.**

| Patient characteristics | AML (TCGA dataset) | | |
|---|---|---|---|
| | **High *CXCL12* (*n* = 86)** | **Low *CXCL12* (*n* = 87)** | ***P*** |
| Age, years | | | 0.537 |
| Median | 60 | 57 | |
| Range | 21–88 | 18–81 | |
| Sex (male/female) | 52/34 | 40/47 | 0.056 |
| WBC count, ×10⁹/L | | | <0.0001 |
| Median | 6.4 | 42.1 | |
| Range | 0.4–116.2 | 0.9–297.4 | |
| BM blasts, % | | | <0.001 |
| Median | 62 | 77 | |
| Range | 30–100 | 32–100 | |
| PB blasts, % | | | <0.0001 |
| Median | 14 | 52 | |
| Range | 0–97 | 0–98 | |
| Karyotype | | | 0.872 |
| Favorable | 15 | 17 | |
| Intermediate | 50 | 51 | |
| Adverse | 20 | 17 | |
| Unknown | 1 | 2 | |
| *FLT3-ITD* | | | 0.010 |
| Present | 7 | 19 | |
| Absent | 79 | 66 | |
| *NPM1* | | | 0.015 |
| Mutated | 17 | 31 | |
| Wild-type | 69 | 54 | |
| *CEBPA* | | | 0.318 |
| Single mutated | 3 | 5 | |
| Double mutated | 4 | 1 | |
| Wild-type | 79 | 79 | |
| *IDH1* | | | 0.980 |
| Mutated | 8 | 8 | |
| Wild-type | 78 | 77 | |
| *IDH2* | | | 0.818 |
| Mutated | 9 | 8 | |
| Wild-type | 77 | 77 | |
| *RUNX1* | | | 0.062 |
| Mutated | 11 | 4 | |
| Wild-type | 75 | 81 | |
| *DNMT3A* | | | 0.965 |
| Mutated | 21 | 21 | |
| Wild-type | 65 | 64 | |
| *TP53* | | | 0.046 |
| Mutated | 10 | 3 | |
| Wild-type | 76 | 82 | |

**Note:**
 Abbreviations: AML, acute myeloid leukemia; TCGA, The Cancer Genome Atlas; WBC, white blood cells; BM, bone marrow; PB, peripheral blood; *ITD*, internal tandem duplication.

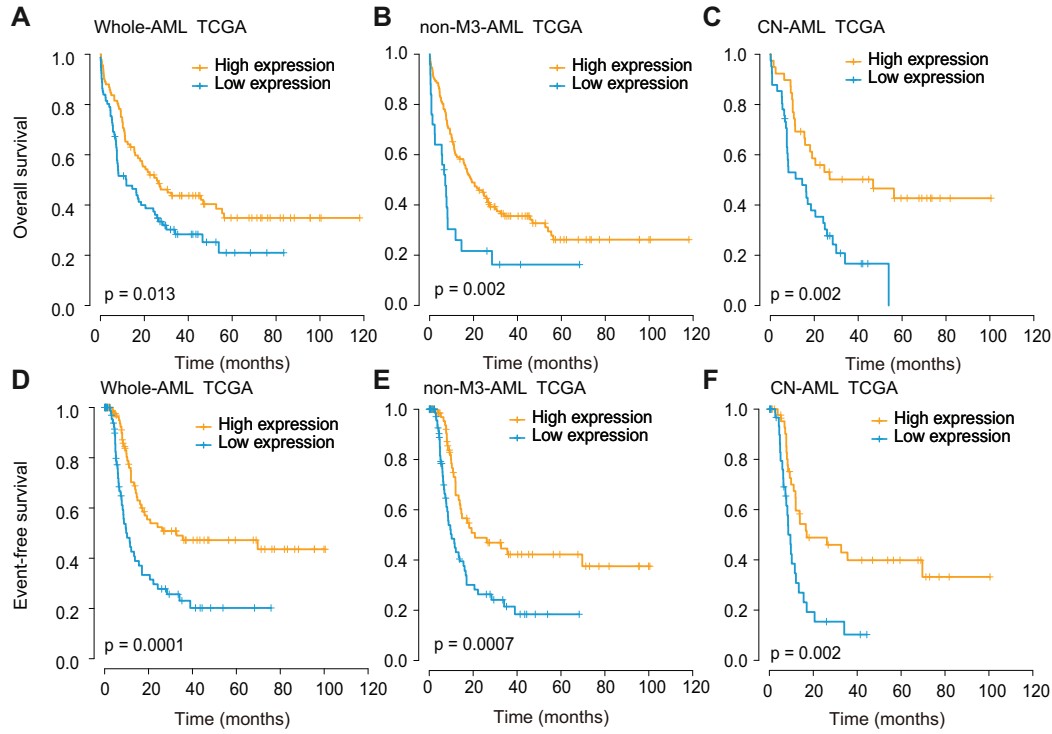

**Figure 4 Survival analysis of *CXCL12* using the TCGA cohort.** (A, D) The correlation of *CXCL12* expression level with OS (A) and EFS (D) for the total AML patients. (B, E) The correlation of *CXCL12* expression level with OS (B) and EFS (E) for the non-M3-AML patients. (C, F) The correlation of *CXCL12* expression level with OS (C) and EFS (F) for the CN-AML patients.

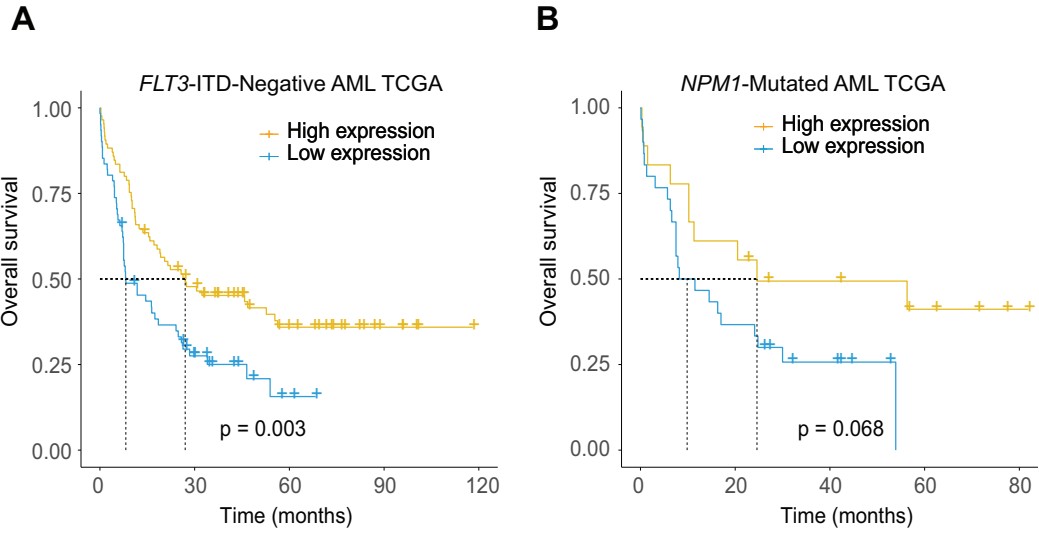

**Figure 5 Prognostic significance of *CXCL12* in the molecularly defined subgroups in the TCGA cohort.** (A) Kaplan–Meier estimate for OS of *FLT3*-ITD-negative AML patients. (B) Kaplan–Meier estimate for OS of *NPM1*-mutated AML patients.

**Table 2 Multivariate analysis of *CXCL12* expression for overall survival in the TCGA cohorts.**

| Variables | Whole-AML ($n$ = 173) | | CN-AML ($n$ = 80) | |
|---|---|---|---|---|
| | Hazard Ratio (95% CI) | $P$ | Hazard Ratio (95% CI) | $P$ |
| *CXCL12*[a] | 2.25 [1.51–3.35] | <0.0001 | 2.14 [1.15–3.97] | 0.016 |
| Age[b] | 1.01 [1.00–1.03] | <0.0001 | 2.37 [1.35–4.18] | 0.003 |
| WBC count[c] | 3.14 [1.92–5.14] | <0.0001 | 1.21 [0.67–2.19] | 0.520 |
| Cytogenetic risk[d] | 1.79 [1.25–2.57] | 0.002 | – | – |
| *TP53*[e] | 2.85 [1.41–5.75] | 0.004 | – | – |
| *RUNX1*[e] | 2.08 [1.10–3.93] | 0.023 | – | – |
| *DNMT3A*[e] | 1.74 [1.13–2.68] | 0.012 | 1.95 [1.11–3.44] | 0.020 |
| *IDH1*[e] | – | – | 0.60 [0.21–1.75] | 0.351 |

**Notes:**
[a] Low vs high expression.
[b] >60 vs ≤60 years.
[c] ≥30 vs <30 × $10^9$/L.
[d] Adverse vs intermediate vs favorable.
[e] Mutated vs wild type.
Abbreviations: TCGA, The Cancer Genome Atlas; CN-AML, cytogenetically normal AML; WBC, white blood cells; CI, confidence interval.
Hazard Ratio > 1 or Hazard Ratio < 1 indicate a higher or lower risk. Only variables with a univariable $P \leq 0.20$ were included in the multivariable models.

activated, mast cells activated, and eosinophils were significantly changed between high *CXCL12* and low *CXCL12* groups, while B cell naive, T cells CD8, T cells CD4 naive, T cells CD4 memory resting, T cells CD4 memory activated, T cell follicular helper, NK cells resting, monocyte, macrophages M0, macrophages M1, macrophages M2, dendritic cells resting, dendritic cells activated, mast cells resting, and neutrophils were not obviously altered between groups. More specifically, high *CXCL12* expressed groups tended to harbor more memory B cells and plasma cells infiltration while low *CXCL12* expressed groups exhibited more eosinophils infiltration. Less importantly, T cells regulatory (Tregs), T cells gamma delta, and NK cells were rarely infiltrated in both high and low *CXCL12* expressed groups.

To verify the signatures of immune infiltration, we adopted multiple deconvolution methods including TIMER, quanTIseq, xCell, MCP-counter, and EPIC to cross-check the analytic results of CIBERSORT. Then, we made a Pearson correlation analysis between all infiltration results and *CXCL12* expression, and totally 16 significantly correlated results were displayed in Fig. S6. Plasma cells (xCell, Cor = 0.273, $P$ < 0.001) and B cells (quanTIseq, Cor = 0.246, $P$ < 0.001; EPIC, Cor = 0.198, $P$ = 0.007; MCP-counter, Cor = 0.145, $P$ = 0.050) had a positive correlation with *CXCL12* expression.

## GO analysis and KEGG pathways analysis of *CXCL12*

GO analyses of *CXCL12* for biological process (BP), cellular component (CC) and molecular function (MF) were performed on the online website of STRING (http://string-db.org) (Fig. 7), using genes differentially expressed between high and low *CXCL12* expressers (|log FC| > 1.5, adjusted $p$-value < 0.05, Supplemental Data 6). GO term annotation showed that *CXCL12* was correlated with anatomical structure morphogenesis, extracellular structure organization, animal organ morphogenesis, biological adhesion,
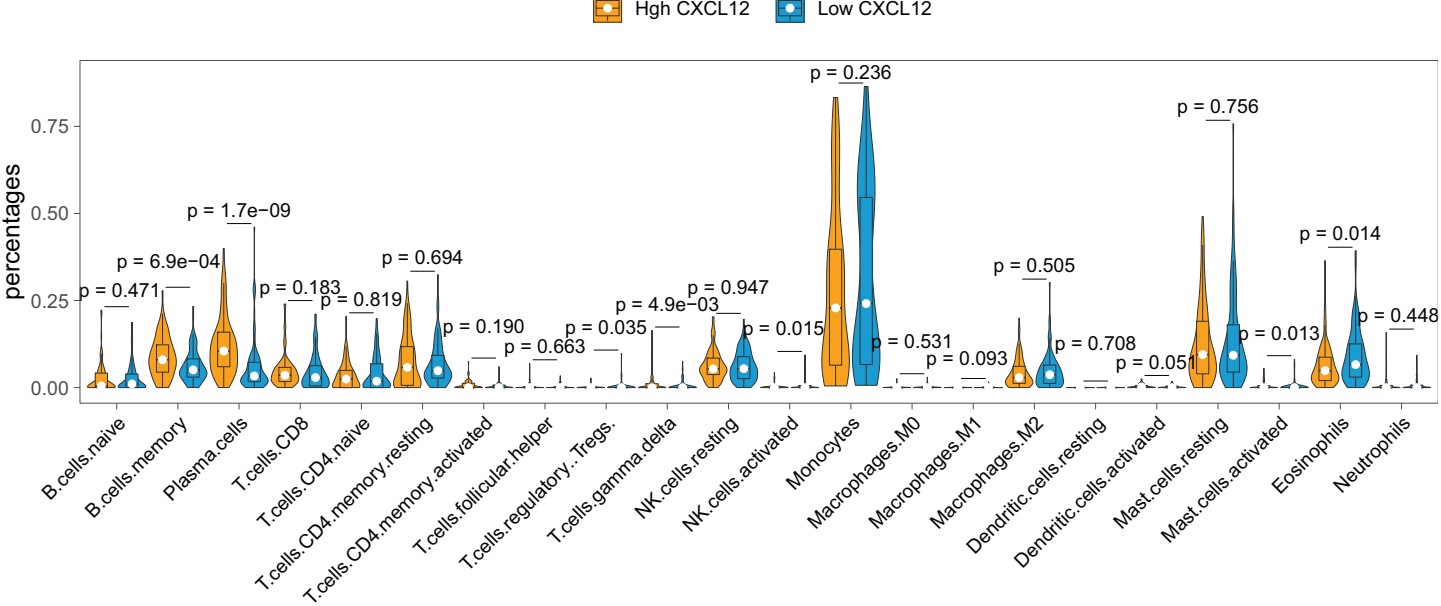

**Figure 6 Immune cell infiltration difference between high and low *CXCL12* expressers.** The Violin plot exhibits the difference between 22 CIBERSORT immune cell fractions between high and low *CXCL12* expressers.

system development, multicellular organismal process, animal organ development, cell adhesion, extracellular matrix organization, and multicellular organism development (BP); extracellular region, extracellular space, extracellular matrix, intrinsic component of plasma membrane, integral component of plasma membrane, cell periphery, plasma membrane, collagen-containing extracellular matrix, basement membrane, and collagen trimer (CC); binding, growth factor binding, signaling receptor binding, calcium ion binding, cell adhesion molecule binding, glycosaminoglycan binding, chemokine activity, heparin binding, receptor ligand activity, and receptor regulator activity (MF) (Fig. 7, Supplemental Data 7). GO analysis showed significantly high enrichment in anatomical structure morphogenesis for biological process, extracellular region for cellular component, and binding for molecular function. Moreover, *CXCL12* was involved directly or indirectly in the protein digestion and absorption pathway according to the KEGG pathways analysis (Fig. 7, Supplemental Data 7).

## DISCUSSION

Our study showed *CXCL12* expression was extremely low in blood and bone marrow compared with other normal tissues and also downgraded in hematological neoplasms among pan-carcinomas. For our purpose, we then validated the expression differences between AML and their normal contrasts in two independent cohorts and found a consistent result. Herein, we concluded that *CXCL12* was notably down-regulated in AML. Hitherto, there hasn't been enough researches to investigate *CXCL12* expression in AML, although its receptor *CXCR4* has been demonstrated to be overexpressed in AML (*Rombouts et al., 2004*). To our knowledge, *CXCL12* plays a prominent role in the interaction between AML cells and the BM microenvironment, and the *CXCL12/CXCR4-*

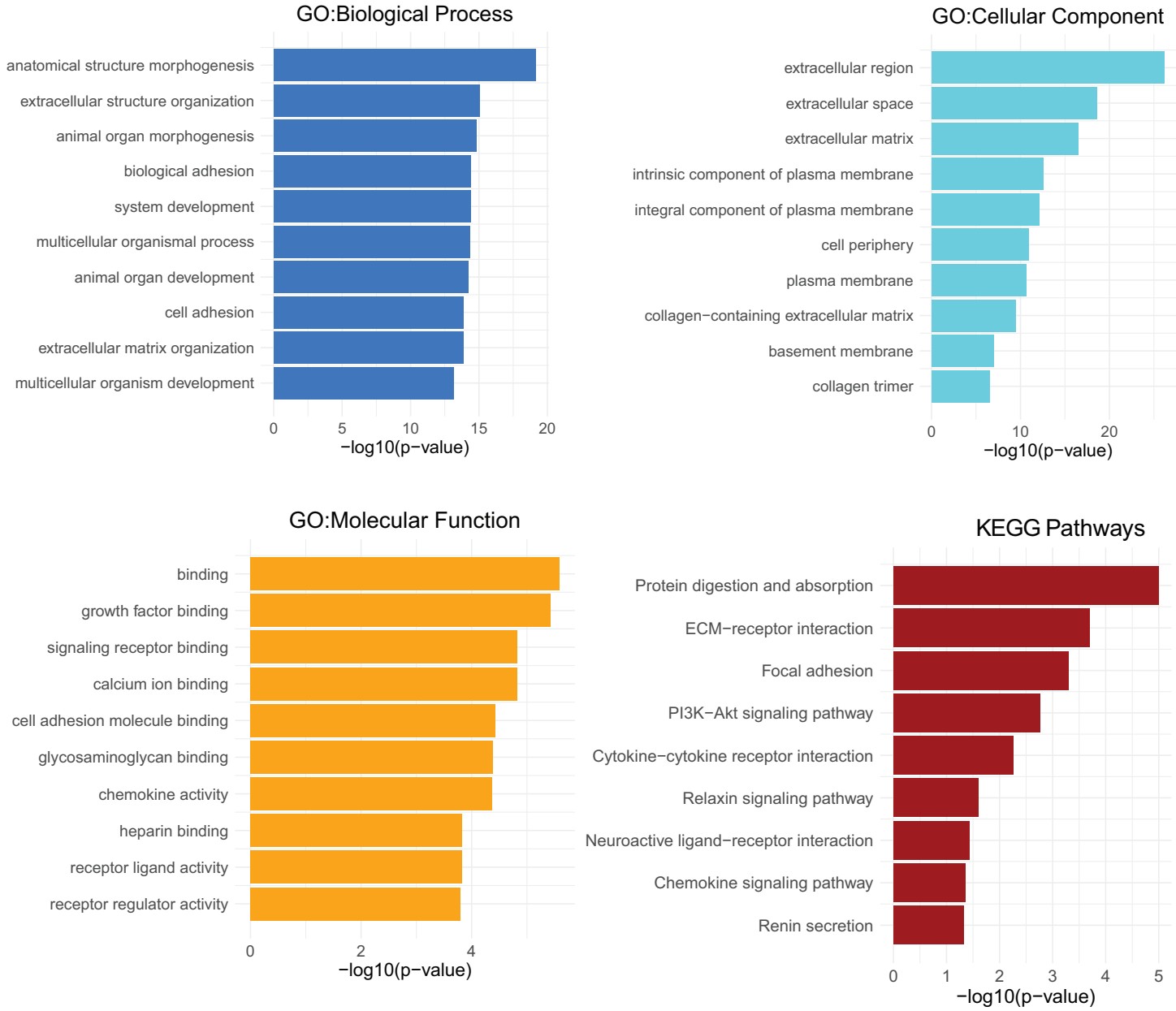

**Figure 7 GO and KEGG pathways analysis of differentially expressed genes between high and low *CXCL12* expressers.** The top-ten statistically significant results identified including biological process (BP), cellular component (CC), molecular function (MF), and KEGG pathways are listed according to their -log10 (*p*-value) (colored bars).

axis is importantly involved in the migration, infiltration and overproliferation of AML cells (*Ladikou et al., 2020*).

The clinical implications of *CXCL12* were further investigated. *CXCL12*low patients tended to have higher WBC counts and more percentages of blasts, implying more active proliferation of *CXCL12*-reduced AML cells. Moreover, *CXCL12*low patients were more likely to harbor *FLT*-ITD and *NPM1* mutations and less often to have *TP53* mutation.

Concerning prognostic relevance, we identified and validated low *CXCL12* expression as a prognostic indicator for inferior OS and EFS in all AML, non-M3-AML, and CN-AML patients. In fact, high *CXCR4* expression level has been associated with inferior prognosis of AML (*Burger & Bürkle, 2007*; *Spoo et al., 2007*) and *CXCR4* antagonists such as AMD3100 have been proven to enhance chemosensitization of AML (*Nervi et al., 2009*). However, it was interesting to discover the predictive value of *CXCL12* for AML prognosis exactly opposite to *CXCR4*, which was beyond our expectation. Previously, Petit I et al. demonstrated that up-regulation of *CXCR4* could serve to increase the sensitivity of cells to lower *CXCL12* signals (*Petit et al., 2002*), and Mandawat et al. found that exposure to *CXCL12* caused the lowering of the surface expression of *CXCR4* on AML cells (*Mandawat et al., 2010*), which might provide an explanation for this phenomenon. Moreover, we pointed out that *FLT3*-ITD absent patients and *NPM1* mutated patients could be further subdivided into high-risk groups (with *CXCL12* down-regulation) and low-risk groups (with *CXCL12* up-regulation) respectively based on the expression level of *CXCL12*.

GO analysis helped us understand the concrete functions of *CXCL12* and the result showed that the gene dynamically participated in anatomical structure morphogenesis for biological process, extracellular region for cellular component, and binding for molecular function. It was also reasonable to extrapolate protein digestion and absorption pathway was curbed molecularly in some way during the development of AML. It has been widely evidenced that *CXCL12* plays an essential role in stem cell anchorage to the BM microenvironment and reduction of *CXCL12* concentrations within the BM may interfere with retention and facilitate the egress of cells (*Petit et al., 2002*), which could promote the progress of AML. As previously demonstrated, *CXCL12* promoted glycolytic reprogramming in AML cells (*Braun et al., 2016*) whereas the malignant cells could benefit from aerobic glycometabolism, which endowed them with better capacity for cell proliferation, immune evasion, and chemotherapeutics resistance (*Vander Heiden, Cantley & Thompson, 2009*). Low *CXCL12* expression restrained glycolytic reprogramming and indirectly promoted aerobic glycometabolism might become one of the possible mechanisms for low *CXCL12* expression inducing poor AML prognosis. When analyzing the relationship between *CXCL12* and immune cell infiltration, we found high *CXCL12* expressed groups tended to harbor more memory B cells and plasma cells infiltration while low *CXCL12* expressed groups exhibited more eosinophils infiltration. However, the pathological role of *CXCL12* in AML is still largely unknown and needs to be elucidated in future studies.

Although we had been committing ourselves to make the research as comprehensive and credible as possible, there were still several drawbacks that couldn't be ignored. Firstly, the collections were entirely acquired from the public database due to the quantity of clinical data was too limited to conduct such a large-scale analysis. We attempted to mend it by analyzing multi-group independent datasets to guarantee our results convincing. And secondly, we did not carry out subsequent experiments to authenticate our results, so more laboratory proofs such as protein level verification are required to be offered for our research.

## CONCLUSIONS

In summary, we carry out a large-scale data analysis of the potential relationship between *CXCL12* and AML for the first time. *CXCL12* is significantly down-regulated in AML and low *CXCL12* expression is an independent and poor predictor of clinical outcome of AML. The gene could provide additional prognostic value in *FLT3*-ITD absent or *NPM1* mutated AML patients. Furthermore, our research replenishes the understanding of the immune and clinical association between *CXCL12* and AML, which could help clinicians to explore new targeted drugs and immunotherapy for AML. Finally, further experiments are warranted to support our findings.

### Funding

This study was supported by the National Natural Science foundation of China (81970118, 81900163), the Medical Innovation Team of Jiangsu Province (CXTDB2017002), the Zhenjiang Clinical Research Center of Hematology (SS2018009), the Social Development Foundation of Zhenjiang (SH2019065), and the Scientific Research Project of The Fifth 169 Project of Zhenjiang (21). There was no additional external funding received for this study. The funders had no role in study design, data collection and analysis, decision to publish, or preparation of the manuscript.

### Grant Disclosures

The following grant information was disclosed by the authors:
National Natural Science foundation of China: 81970118 and 81900163.
Medical Innovation Team of Jiangsu Province: CXTDB2017002.
Zhenjiang Clinical Research Center of Hematology: SS2018009.
Social Development Foundation of Zhenjiang: SH2019065.
Scientific Research Project of The Fifth 169 Project of Zhenjiang (21).

### Competing Interests

The authors declare that they have no competing interests.

### Author Contributions

- Shi-Sen Wang performed the experiments, analyzed the data, prepared figures and/or tables, and approved the final draft.
- Zi-Jun Xu performed the experiments, analyzed the data, prepared figures and/or tables, and approved the final draft.
- Ye Jin performed the experiments, analyzed the data, prepared figures and/or tables, and approved the final draft.
- Ji-Chun Ma performed the experiments, authored or reviewed drafts of the paper, and approved the final draft.
- Pei-Hui Xia analyzed the data, authored or reviewed drafts of the paper, and approved the final draft.
- Xiangmei Wen analyzed the data, authored or reviewed drafts of the paper, and approved the final draft.
- Zhen-Wei Mao conceived and designed the experiments, authored or reviewed drafts of the paper, and approved the final draft.
- Jiang Lin conceived and designed the experiments, authored or reviewed drafts of the paper, and approved the final draft.
- Jun Qian conceived and designed the experiments, authored or reviewed drafts of the paper, and approved the final draft.

## Data Availability

The datasets analyzed in this study are available in the following open access repositories:

The Cancer Genome Atlas (TCGA), https://cancergenome.nih.gov/.

Genotype-Tissue Expression (GTEx), https://gtexportal.org/.

Cancer Cell Line Encyclopedia (CCLE), https://portals.broadinstitute.org/ccle/.

Gene Expression Omnibus (GEO): GSE6891, GSE10358, GSE30029, GSE63270, GSE42519, GSE116256.

The raw measurements are available in the Supplemental Files.

## Supplemental Information

Supplemental information for this article can be found online at http://dx.doi.org/10.7717/peerj.11820#supplemental-information.

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
