# Peer review of "Clinical and prognostic relevance of CXCL12 expression in acute myeloid leukemia"

_PeerJ, doi:10.7717/peerj.11820_

## Round 0.1 · original submission · Major Revisions

Please consider all points raised by the referees.

·

Basic reporting

This is a well written paper dealing with an important issue in the field of acute myeloid leukemia diagnosis and treatment. I believe this paper is offering a relevant contribution to the knowledge of the disease. It is clear and not ambigous. Easy to be read and understood. I do not have major comments, just from the ending point of view:
Article meets the required targets. Rerences are more than suffient ( can be reduced). Too many figures.

Experimental design

Very well desuigned retrospective registry study

Validity of the findings

The paper is too long and can be significantly reduced without losing an important message.
This is a retrospective research article based on different data base screening and analysis. Because this is not a prospective research study conclusion must be more prudent limiting to suggest prospective controlled studies. Authors correctly underlined this limitation in the last discussion paragraphs but I believe limitation should be reported in the abstracts also. Moreover I suggest a more prudent title.
Too many figures.
.

Additional comments

I support pubblication after editing revision

Reviewer 2 ·

Basic reporting

no comment

Experimental design

As a computational analysis study, this manuscript directly goes to the single CXCL12 gene without giving a systematic overview of differential genes or cytokines in AML, so there is weakness in experimental design before diving into this particular cytokine.

Validity of the findings

no comment.

Additional comments

This study explored the prognostic value of CXCL12 in acute myeloid leukemia by associating CXCL12 expression with immune infiltration, pathway enrichment, and cancer survival. The exploration is generally valuable analysis, however, there are multiple concerns as following.
- The decision to focus on specifically CXCL12 is not clear. The authors can improve by at least starting looking at genes/cytokines most differential between AML and control to provide the reason to focus on this specific cytokine, rather than cherry-picking one cytokine.
- The authors used CIBERSORT for immune deconvolution; due to the high heterogeneity of immune infiltration deconvolution algorithms with different cell types available, a better practice is to utilize multiple deconvolution methods to cross-check the association with immune infiltrates. For example, popular deconvolution methods include TIMER, CIBERSORT, quanTIseq, xCell, MCP-counter, EPIC, and immuneDeconv. These methods can be used to validate whether the significant B cell difference is reproducible for CXCL12 expression. In fact, the CIBERSORT algorithm is incorrectly written as CIBERSOFT in line 130.
- The unit of gene expression from GTEx and any transformation shall be noted, e.g. in Fig1a whether this is TPM or log scale. Same issue in Fig1b for CCLE expression unit, and also for all following figures.
- In comparison with the hematopoietic stem cells of differentiated states, there shall be statistics and number of samples shown to prove the higher expression of CXCL12 in HSC than the differentiated states.
- Given that CXCL12 in the current manuscript likely link to monocytes and some other immune cells, it is suggested that single-cell data can be used to show the cell-type resource of CXCL12, and study the potential interaction with other cell types based on cytokine-receptor interaction.
- There are analyses of mutation status with possible synergy with CXCL12. There shall be mechanistic research into why the different mutations can be related to the CXCL12 level.

---

## Round 0.2 · accepted · Accept

The authors have adequately addressed the issues raised by the referees.